

# Exploring the potential of deep eutectic solvents (DES) in bioactive natural product research: from DES to NaDES, THEDES, and beyond

Mariana Ruesgas Ramon[1], Erwann Durand[2,3], Karlina Garcia-Sosa[1] and Luis Manuel Peña-Rodríguez[1]

[1] Biotechnology Unit, Centro de Investigación Científica de Yucatán, Mérida, Yucatán, México
[2] Qualisud, Univ Montpellier, Avignon Université, CIRAD, Institut Agro, IRD, Universite de La Reunion, Montpellier, France
[3] CIRAD, UMR Qualisud, Montpellier, France

Corresponding author
Mariana Ruesgas Ramon,
azerbet@hotmail.com

## ABSTRACT

**Background:** Deep eutectic solvents (DESs) have garnered significant interest in natural products research, owing to their green and natural attributes in comparison to conventional solvents. However, the approach of demonstrating that DESs are superior extractants has led to an underestimation of their full potential in this field. This perspective disregards crucial challenges related to their practical application and potential scalability, mainly the difficulty of target component separation from intermolecular network forming by DESs. Conversely, the DESs unique features can enhance aspects such as solubilization, stabilization, and storage of natural products, as well as improve their biological activities. By addressing key challenges and limitations, we aim to provide valuable insights into the potential of DESs in this domain.

**Methodology:** In this review, we conducted an exhaustive literature search to gather relevant articles about DESs and their applications in bioactive natural product research. The gathered literature was analyzed, and a systematic thematic categorization was performed, emphasizing studies where the use of DESs yielded relevant outcomes that could potentially present an advantage in the exploration of bioactive natural products. The approach in structuring this review aimed to provide an overview of the potential and challenges associated with DESs in the domain of bioactive natural product research, transcending their conventional role as mere extraction solvents.

**Results:** Through this critical analysis of the literature, this review delves into the potential of DESs as effective solvents for the solubilization, stabilization, and storage of bioactive natural products. In addition, it highlights the ability of DESs to improve the biological activities of natural products, as well as to be used as formulation media for the transport of pharmaceutical active ingredients. By revealing these advances, the review contributes to a more complete understanding of DESs and its applicability in the field of bioactive natural products research.

**Conclusions:** The studies compiled in this review underscore the expanded potential of DESs, beyond extraction, finding relevance in the realm of natural products
research. Notably, they contribute to enhancing the desired attributes of the final product, signifying a promising avenue for future advancements in this field.

## INTRODUCTION

The study of bioactive natural products (NP) has played a prominent role in human history, determining modern society's success. NP have been useful for thousands of years as healing agents and new drugs (*Bernardini et al., 2018*). Additionally, their diverse biological effects have proved to be useful in medicine and diverse fields, such as flavoring and preservation of food, cosmetology, and the textile industry (*Croteau, Kutchan & Lewis, 2000*).

The NP research encompasses a series of steps that goes from their identification and characterization to their application in diverse fields. During the process, it is crucial to consider several factor to guarantee optimal performance. Among them, solvent selection is usually crucial to achieving the complete extraction and solubilization and the stabilization of NP of interest (*Kist et al., 2021*). Furthermore, at present, factors such as cost, safety, and sustainability of solvent are of utmost importance, to meet the established guidelines in the Twelve Principles of Green Chemistry (*Anastas & Eghbali, 2010*).

From this standpoint, the interest of the scientific community for the applications of deep eutectic solvents (DESs), and natural deep eutectic solvents (NADES) in NP research. This heightened interest can be primarily attributed to their interesting and modifiable physicochemical attributes (*Shishov et al., 2020*; *Ramón & Guillena, 2019*; *Hansen et al., 2020*). In addition, these features opens the possibility of designing them to meet the experimental expectations, which also can be adjust with the green chemistry trend as well. Thus, they are considered as an excellent alternative to replace conventional solvents commonly used in the extraction stage (*Meng et al., 2018*; *Skarpalezos & Detsi, 2019*; *Su et al., 2017*), of different biomolecules (*e.g.*, phenolic compounds, terpenoids, alkaloids, among others), from plant sources (*Ali Redha, 2021*; *Ruesgas-Ramon, Figueroa-Espinoza & Durand, 2017*; *Zhang et al., 2023*).

However, it is important to mention that although the ability of DESs and NADES to solubilize different types of molecules is remarkable; their use as extraction solvents is limited, especially in industrial applications. One of the main reasons is the difficulty of recovering the NP of interest, mainly due to the very low vapor pressure and relatively high viscosity of NADES (*Socas-Rodríguez et al., 2021*; *Ivanović, Razboršek & Kolar, 2020*). To address this challenge, the use of resins and/or elution/precipitation strategies with conventional solvents have been proposed, but they are limited and cannot be generalized (*Zainal-Abidin et al., 2017*). Recently, a novel approach called switchable-hydrophilicity NADES-based system, which consists of a switchable solvent with the reversible transformation between hydrophilicity and hydrophobicity. This technique offers the

possibility of separating the compound of interest or fractionating it from the solvent (*Sed et al., 2018*). However, to the best of our knowledge, this technique has been tested using hydrophobic NADES-based on fatty acids, in order to recover hydrophobic molecules as well. As a result, it remains a limited technique with certain constraints.

Currently, numerous reviews delve into the use of DESs and NADES within NP research, with a focus on aspects such as enhanced yields and eco-friendly characteristics (*Gullón et al., 2020*; *Ali Redha, 2021*; *Zhang et al., 2022*; *Hikmawanti et al., 2021*; *Suthar et al., 2023*). However, they continue to highlight the approach of their use as extraction solvents that present advantages over those that are conventionally used. This approach not only underestimates the full potential of DESs and NADES but also fails to recognize the ultimate objectives within the realm of NP research.

Therefore, the rationale for this review is described below:

This review is imperative in light of the growing interest in utilizing DESs and NADES in NP research. While existing reviews primarily emphasize their extraction capabilities, this review addresses their utilization from a broader potential perspective. By compiling findings that transcend the extraction, this manuscript showcases the versatility of DESs and NADES, encouraging researchers to explore their novel applications. Given the increasing focus on sustainable and innovative solutions in the scientific community, this review bridges the gap between the narrow extraction-centric perspective and the manifold possibilities these solvents offer.

On the other hand, this review is targeted to researchers, scientists, and professionals actively involved in NP research, innovation technologies, bio-formulations, and related fields. It will serve as a valuable resource for those seeking to diversify their understanding of DESs and NADES, moving beyond their traditional applications. Furthermore, this review will be useful with individuals interested in sustainable chemistry and the development to innovative solutions for industrial and biomedical challenges. Its insights inform about current advancements and opens the door to new research opportunities, making it relevant across academic and industry circles.

## SURVEY METHODOLOGY

This section describes the methodology followed in the writing of this manuscript:

- *Selection of topic*: This review's topic aims to explore how the distinct features of DESs can enhance the effectiveness of bioactive natural products
- *Define the scope*: At this point, it clearly defined the boundaries and scope of this review. That included the DESs application in bioactive natural product research divided into solubilization, stabilization, formulation media, and drug delivery to maintain logical coherence throughout the review.
- *Extensive literature review*: a literature search was conducted thoroughly across reputable scientific databases, including PubMed, Web of Science, Scopus, and pertinent academic journals keywords such as biomolecules, natural products, deep eutectic solvents, formulation, stabilization, solubilization, pharmaceuticals, and storage were

included. Besides, the studies published between 2003 and 2013 were considered to ensure the capture of the most relevant and recent studies in the field.

- **Data extraction and analysis**: Extracted data encompassed an array of crucial parameters, including the specific types of DESs utilized, the bioactive natural products studied, the wide-ranging applications investigated, the methodologies employed, and the resulting outcomes. To ensure a neutral and unbiased approach, a systematic method was adopted for data extraction.

- **Thematic organization**: The collected literature was thoughtfully organized thematically, with a strategic focus on highlighting the diverse facets of DESs applications. This organization encompassed distinct topics, ranging from solubilization and stabilization to formulation media and drug delivery. The structural approach aimed to offer readers a comprehensive panorama of both the potentials and the challenges associated with DESs implementation in the realm of bioactive natural product research.

- **Analyze and interpret the results**: Objective comparison and synthesis of findings from different studies were used to identify trends, disparities, and consensus in DESs applications in natural product research. Conflicting results were analyzed to present an unbiased overview of the current state of knowledge.

- **Rigorous peer review process**: The manuscript underwent a rigorous peer review by subject matter experts, ensuring that the content met the highest standards of scientific rigor, impartiality, and accuracy. Feedback received was thoughtfully incorporated to refine and enhance the manuscript's overall quality and objectivity.

- **Finalization and writing**: The manuscript was written and revised based on feedback received, in order to ensure consistent structure, coherence, and logical flow throughout the document.

- **Citation and referencing**: All the information incorporated was accurately cited, and referenced throughout the manuscript, maintaining a uniform citation style.

- **Proofreading and editing**: All document was carefully proofread the review for grammatical errors, typos, and formatting issues.

- **Submission and publication**: The manuscript was prepared for its submission following the submission guidelines and requirements of the chosen publication.

## DEEP EUTECTIC SOLVENTS: OVERVIEW

After the birth of the concept of Green Chemistry in the early 1990s, defined as the design of chemical products and processes to reduce or eliminate the use and generation of hazardous substances (*Anastas & Eghbali, 2010*), scientific and industrial research has been seeking new alternatives to minimize their impact on the environment. Although the best solution would be to implement solvent-free processes, using solvents is almost unavoidable because of their crucial role in dissolving solids, promoting or allowing mass and heat transfer during reactions, and carrying out the separation and purification of products, among other applications.

In response to this scenario, DESs were developed in 2003 as analogs of IL (ionic liquids), but with two main differences: the source of their starting materials and, to a lesser extent, a simpler preparation. At that time, the DESs concept was coined to describe mixtures of amides with quaternary ammonium salts that had melting points much lower than those of their pure compounds (*Durand et al., 2021*).

Currently, there is no precise or universal definition of DESs, as some of the features used to describe them (*i.e.*, eutectics, hydrogen bond complexes, green, natural), do not universally apply to all cases. For instance, the eutectic point is not exclusive to DESs as it is essentially present in all mixtures of compounds that are completely or partly immiscible in the solid phase. Additionally, there exist DESs (*e.g.*, NADES) that can form without reaching the eutectic point (*Durand et al., 2021*). Similarly, hydrogen bonds can also exist between two components in a mixture *e.g.*, mixtures of fatty acids or alcohols, which cannot be considered DESs (*Martins, Pinho & Coutinho, 2019*; *Paiva, Matias & Duarte, 2018*). On the other hand, there is the supposed "green" nature of DES, a key characteristic that has led to an increase in their use in the investigation of NP. However, the greenness of DESs is determined by the starting materials used as well as the industrial process involved to obtain them. For example, it is often argued that the choline used in the preparation of DESs or NADES is natural (*e.g.*, a group of B vitamins) whereas industrially, it is produced from a reaction involving ethylene oxide, methanol (MeOH), trimethylamine, and hydrochloric acid, and it is cheap primarily because it is produced at a large scale. In addition, the "natural" criterion does not guarantee an absence of toxicity or dangerousness for humans. Therefore, instead of solely emphasizing the green or natural aspects of DESs and NADES, a more comprehensive approach should consider various factors such as their availability, price, recyclability, synthesis, toxicity, biodegradability, performance, stability, flammability, storage, and renewability. Furthermore, it is essential to assess their capacity to improve the development of processes or product formulations.

A more accurate definition of the DESs, in addition to the aforementioned aspects, would be the association of two or more pure components in a defined molar ratio, which can establish non-covalent intermolecular bonds resulting in a stable room temperature liquid phase (*Huang et al., 2019*). Aditionally, the unique properties of these media originate from the short-range structural solvation and the unusual dynamics of species in solution (*Abbott, Edler & Page, 2021*).

The intermolecular interactions present in DESs, mostly through hydrogen bonding, occasional electrostatic forces, and Van der Waals interactions, result in a melting temperature far below those of the individual components (*Zhang et al., 2012*). Typically, these non-covalent bonds between the components in DESs depend on their chemical nature, playing the roles of hydrogen bond donor (HBD) or hydrogen bond acceptor (HBA) (Fig. 1).

Since the first description of DESs in 2003 (*Abbott et al., 2003*), a large number of different components have been used in their formulation. Currently, the most popular DESs formulations include choline chloride (ChCl) in combination with urea, ethylene glycol, and glycerol, as well as other alcohols, amino acids, carboxylic acids, and sugars (*Ruesgas-Ramon, Figueroa-Espinoza & Durand, 2017*).
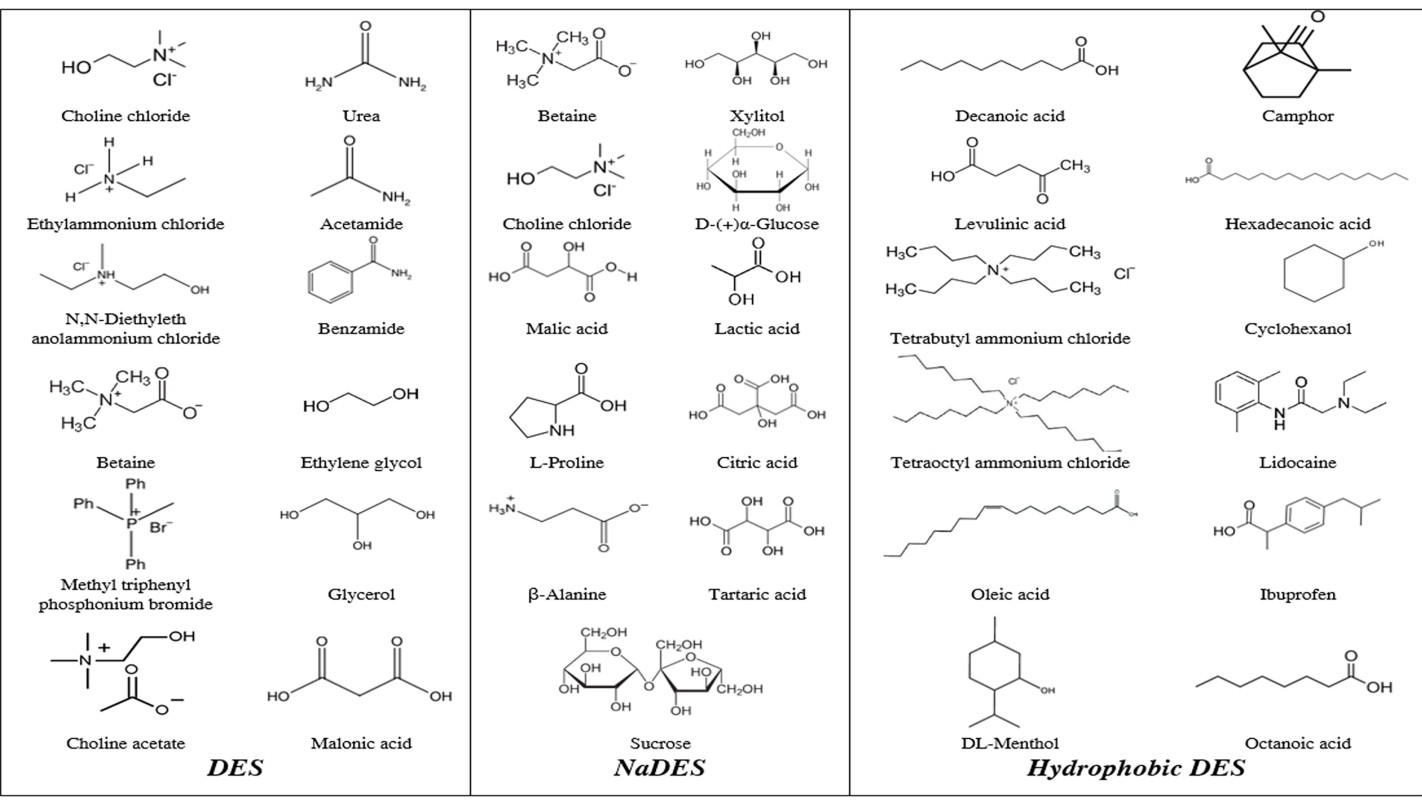

**Figure 1  Commonly used components in the preparation of DES, NaDES, and hydrophobic DES.**

To date, different methodologies have been reported for the preparation of DESs (Table 1); some of these methodologies could be tailored (*e.g.*, modifying the time of heating or temperature) to fit the nature of the starting components. The versatility in the synthesis of DESs through different methodologies bestows the significant advantage of simplicity and accessibility in the formulation process, eliminating the need for specialized equipment like grinding or elaborate heating and stirring apparatus. Furthermore, the typically biodegradable and low-toxicity nature of DESs components obviates the necessity for post-synthesis purification steps (*Abbott et al., 2003*; *Radošević et al., 2016*).

Just as the different synthetic methodologies, DESs classification, and nomenclature have evolved since first described in 2013. The first nomenclature, proposed in 2007 (*Abbott, Harris & Ryder, 2007*), classified DESs according to the nature of the complexing agent used, describing them by the formula $Cat^+X^-zY$, where $Cat^+$ is any ammonium, phosphonium, or sulfonium cation, and $X$ is a Lewis base, generally a halide anion. The complex anionic species are formed between $X^-$ and either a Lewis or Brønsted acid $Y$ ($z$ refers to the number of Y molecules that interact with the anion). On this basis, DESs have been classified into four different types (Table 2) (*Ramón & Guillena, 2019*; *Smith, Abbott & Ryder, 2014*). Among them, type III have proved to be the most agreeable to the green chemistry concept, because of their simple preparation, high biodegradability, low cost of starting materials, the versatility of main ingredients, nonreactivity with water, *etc.*

**Table 1 Examples of used methods for deep eutectic solvents synthesis.**

| Synthesis method | Operation conditions | Equipment or materials | Reference |
|---|---|---|---|
| Heating and stirring | 40–60 °C until a clear liquid is formed (about 2 h) | Magnetic stirrer | *Abbott et al. (2003)*, *Dai et al. (2013)* |
| Grinding | Room temperature until a clear liquid is formed | A mortar with a pestle | *Florindo et al. (2014)* |
| Vacuum evaporation | 50 °C | Rotatory evaporator Desiccator | *Dai et al. (2015)* |
| Freeze drying | −20 °C freeze-dried until a clear viscous liquid is formed (about 24 h) | Freeze-dryer | *Gutiérrez et al. (2009)* |
| Microwave-assisted synthesis | 20 s 200 W | Microwave oven | *Gomez et al. (2018)* |
| Ultrasound-assisted synthesis | 50 °C until a clear liquid is formed (10–15 min) 37 kHz, 30 W | Ultrasonic bath | *Bajkacz & Adamek (2018)*, *Santana et al. (2019)* |
| Mechanochemical | 55–65 °C About 4 min 50 rpm | Twin screw extrusion (TSE) | *Crawford et al. (2016)* |

**Table 2 General formulas of deep eutectic solvents.**

| Types | General formula | Terms | Example |
|---|---|---|---|
| I | $Cat^+X^-zY + xMCl_x$ | M=Zn, In, Sn, Al | $ChCl + ZnCl_2$ |
| II | $Cat^+X^-zY + xMCl_xyH_2O$ | M=Cr, Ni, Cu, Fe | $ChCl + CoCl_2 . 6H_2O$ |
| III | $Cat^+X^-zRZ$ | Z=OH, COOH, CONH$_2$ | $ChCl + urea$ |
| IV | $MCl_x + zRZ$ | M=Zn, Al and Z=OH, CONH$_2$ | $ZnCl + urea$ |

(*Mbous et al., 2017a*). This kind of DESs are typically formed by a quaternary ammonium salt like ChCl, choline acetate, or ethyl ammonium chloride as HBA, and an amine, amide, carboxylic acid, sugar or other polyols as HBD (Fig. 1).

Recently some DESs formulations have been described that do not fit with the aforementioned nomenclature. These include the recently introduced concept of NADES (natural deep eutectic solvents) (*Choi et al., 2011*), which could be considered as a type III DESs. However, NADES refers to the nature of its components and the possibility of its being a third liquid phase in cells of living organisms, as an alternative to water and lipids and thought to perform specific functions. It is important to mention that NADES is often used to refer to DESs formulated using natural components, usually plant primary metabolites (*e.g.*, sugars) (*Benvenutti, Zielinski & Ferreira, 2019*).

Although DESs and NADES share very similar physicochemical properties (strong ability to dissolve protic molecules, low vapor pressure, and miscibility with water, among others) and, in some cases, are referred to indistinctly, they each have special features. The NADES concept involve possible roles in nature, normally they are exclusively formed

by nonionic species (*Ruesgas-Ramon, Figueroa-Espinoza & Durand, 2017*) and, unlike DESs, can be formed without reaching the eutectic point (*Durand et al., 2021*).

Recently, the formulation of a new kind of DESs (type V) was achieved by combining DL-menthol with various quaternary ammonium salts, resulting in the formation of hydrophobic eutectic mixtures designated as Hydrophobic Deep Eutectic Solvents (*Ribeiro et al., 2015*). This was confirmed by the successful formulation of DL-menthol with various organic acids, resulting in the creation of DESs that demonstrated clear immiscibility with water (Fig. 1).

Additional nomenclature related to DESs include the term therapeutic deep eutectic solvents (THEDES), which is used for DESs containing active pharmaceutical ingredients (APIs) as part of their formulation (*Aroso et al., 2016*). More recently, the formulation of new low melting mixtures (LMM) inspired in ChCl: Urea but based on cyclodextrins (CD) and levulinic acid produced mixtures that remained liquid at room temperature, had relatively low viscosity and showed supramolecular properties. This emerging class of DESs capable of solubilizing poorly water-soluble drugs and/or protect fragile molecules during topical cutaneous administration have been designated as SUPRADES (*El-Achkar et al., 2020b*, *2020a*).

The countless combinations that can lead to the formation of different NADES result in versatile solvents with applications across various research fields.

In an effort standardize the nomenclature, in this review, DESs and NADES will be referred to as (Na)DES.

## (Na)DES: BEYOND THEIR USE AS EXTRACTION SOLVENTS

The existing literature extensively demonstrates the (Na)DES's capability to extract bioactive molecules from a wide range of plant materials. However, in this type of research, it is common to follow the same approach as when using conventional solvents, where the ultimate goal is to obtain a pure molecule. Nevertheless, the unique ability of (Na)DES to establish complex molecular interactions with different NP (*Dai et al., 2016*) can be a significant advantage, particularity when the separation and purification of target molecules are not necessary. Recent studies have shown the ability of (Na)DES to enhace the solubility, stabilizaty, and biological properties of NP. Moreover, the use of (Na)DES has also been proposed as storage media for NP (*Jeong et al., 2017*), and as drug delivery vehicles in the development of therapeutic systems (THEDES) (*Kist et al., 2021*; *Hamdi et al., 2019*).

### (Na)DES for the solubilization and stabilization of natural products

The original concept of (Na)DES was used to designate an alternate liquid medium (in addition to water and lipids) present in living organisms, which could play biological and physiological roles. Although these roles have not yet been fully elucidated, some studies have provided supporting the idea that (Na)DES are present in nature. For instance, it has been shown that the consistency and the equimolar ratio between components of some nectars (fructose-glucose-sucrose), honey (glucose and fructose), or maple syrup (sucrose, glucose, fructose, and malic acid) are very similar to those of synthetic (Na)DES (*Choi*

*et al., 2011*). Furthermore, and as an attempt to elucidate the possible role of (Na)DES in nature, the solubility of some NP, including quercetin, cinnamic acid, carthamin, taxol, ginkgolide B, and 1,8-dihydroxyanthaquinone, was evaluated in several (Na)DES formulations. The results revealed that (Na)DES are not only capable of solubilizing NP of a wide range of polarities, but also exhibit solubilization properties 18 to 460,000 times higher than those of water (*Dai et al., 2013*). Subsequently, some reports have described the solubilization ability and stabilization capacity of (Na)DES when used for direct extraction of plant matrices (*Ruesgas-Ramon, Figueroa-Espinoza & Durand, 2017*; *Skarpalezos & Detsi, 2019*).

One of the most interesting cases involves the solvation of rutin, a secondary metabolite that, despite its polyhydroxylated structure, is almost insoluble in water. However, hydrated (Na)DES showed an ability to dissolve rutin that was 50 to 100-fold higher than that of water (*Choi et al., 2011*). These results were confirmed when rutin was extracted from the flowers of *Sophora* japonica L. (Fabaceae) using twenty different hydrated ChCl-based (Na)DES (*Zhao et al., 2015*).

Similar to rutin, carthamin, the primary red colorant found in Safflower (*Carthamus tinctorius* L.), exhibits low solubility in water and is prone to instability in aqueous environments and when exposed to light. Bearing this in mind, a comparative assessment of carthamin's stability in both water and a 40% ethanol (EtOH) (v/v) solution was conducted, as opposed to various (Na)DES, under varying conditions of temperature, light exposure, and duration. These findings revealed that xylitol-ChCl (1:4 molar ratio) outperformed the aqueous medium, providing superior protection against thermal degradation. However, (Na)DES formulations involving sucrose-ChCl (1:4) and glucose-ChCl (2:5) displayed the most effective stabilization effects.

Moreover, it's noteworthy that carthamin maintained its stability in both sucrose-ChCl (1:4) and proline-malic acid (1:1) (Na)DES formulations. However, its stability decreased with the increasing hydration level of the (Na)DES. This observation suggests that the stabilization mechanism is closely related to the molecular structure of the target nanoparticles and the water content within the (Na)DES formulation. Furthermore, the stability of carthamin within viscous, low-water (Na)DES reinforces the hypothesis that (Na)DES have the potential to maintain the stability of nanoparticles within living cells, even in conditions with limited water content (*Solér, Bergström & Shanahan, 2009*).

Nonetheless, it is intriguing to note that these findings do not align with factors typically associated with the solubilization of nanoparticles in (Na)DES. Conventionally, the addition of water reduces viscosity and enhances the solubilization capacity of (Na)DES, which differs from the observed stability of carthamin (*Ruesgas-Ramon, Figueroa-Espinoza & Durand, 2017*).

Similarly, salvianolic acid B (SAB), well-regarded for its anti-inflammatory and neuroprotective properties, is susceptible to instability in aqueous solutions, frequently resulting in a decline in its clinical efficacy. Evaluation of the stability of SAB in four ChCl-based (Na)DES, combined with different HBDs (*e.g.*, ethylene glycol, 1,2-propylene glycol, glycerol, and 1,4-butylene glycol), This evaluation was compared against stability in traditional solvents like MeOH, EtOH, and water at elevated temperatures of 60 and 90°C.

The results demonstrated that SAB maintained higher stability in all (Na)DES systems. Among them, the ChCl-glycerol (1:2) formulation exhibited a notably reduced degradation rate for SAB in contrast to other (Na)DES.

The observed differences in the stability and solvation of SAB within these diverse (Na)DES were elucidated through interactions identified in FT-IR studies (*Chen et al., 2016*).

The anthocyanins are the most important and widespread group of pigments found in plants; like other flavonoids, they exhibit multiple pharmacological activities such as antioxidant, antiatherosclerosis, and antihyperlipidemic, among others. Since anthocyanins are also highly unstable, being particularly susceptible to degradation by factors such as temperature, light, and organic solvents commonly used for their extraction (*Dillard & Bruce German, 2000*), the use of (Na)DES has been proposed as an alternative to their extraction from *Catharanthus roseus* L. (Apocynaceae). The study found that (Na)DES formulations such as 1,2-propanediol–ChCl (1:1) and lactic acid–glucose (5:1) were equally effective in extracting compounds as acidified MeOH. However, when evaluating stability using cyanidin as a model compound, lactic acid–glucose (5:1) (Na)DES showed significantly better anthocyanin stability compared to acidified EtOH, especially at high temperatures (*Dai et al., 2016*). In a similar study, the extraction efficiency and the stability of anthocyanins obtained from mulberry were evaluated using ten different (Na)DES formulations; of these, a formulation of ChCl-lactic acid (1:2) hydrated with 20% (v/v) produced the best yield of extraction and thermal stability of anthocyanidins, when compared to acidified EtOH (*Bi et al., 2020*).

In a recent investigation focused on the stability of anthocyanins extracted from blueberry peels using ChCl-based (Na)DES, it was discovered that ChCl-malic acid (1.5:1), ChCl-citric acid (2:1), and ChCl-lactic acid (1:1) demonstrated comparable anthocyanin stability when compared to the reference hydroalcoholic solution (*Grillo et al., 2020*).

Further examination of ChCl-lactic acid (1:1) (Na)DES, which exhibited the highest extraction yield, was conducted to assess its ability to safeguard anthocyanins from degradation during the extraction process, which involved the use of microwave and ultrasonic irradiation. Surprisingly, this (Na)DES displayed a significant stabilizing effect, even under the harsh conditions of microwave and ultrasonic-assisted extraction. These findings hold particular significance as microwave-assisted extraction (MAE) and ultrasound-assisted extraction (UAE) techniques, although commonly employed to enhance extraction efficiency, can potentially lead to adverse effects when traditional solvents are used.

The superior extraction yields achieved with (Na)DES in conjunction with UAE or MAE can be attributed to the rapid heating properties of these solvents, which reduce viscosity, thus improving mass transfer and resulting in higher extraction yields (*Grillo et al., 2020*).

In the same vein, a series of (Na)DES were tested to extract bioactive compounds from blueberry leaves to add value to this feedstock. The results demonstrated that the combination of lactic acid: sodium acetate: water (3:1:2) and ChCl: oxalic acid (1:1) yielded superior performance in the recovery of phenolic compounds (1.6-fold and 2.2-fold higher efficacy, respectively) compared to MeOH. Additionally, this study confirmed that the

(Na)DES composition plays a key role in the type of biomolecules that can be extracted from the same plant material. In fact, lactic acid-based (Na)DES enabled the extraction of hydroxycinnamic acids and flavonols, whereas the ChCl-based (Na)DES proved to be selective for the extraction of anthocyanin. Interestingly in this case the extraction efficacy for anthocyanins was significantly improved, clearly surpassing that provided by the conventional solvent (*Santos-Martín et al., 2023*).

Like berries, turmeric (*Curcuma longa* L.) has been used for centuries in traditional Asian medicine to treat a variety of ailments, and more recently, it has been considered a superfood (*Sharma et al., 2021*).

Since the clinical applications of curcumin, the principal curcuminoid of turmeric reported to possess antioxidant, anti-inflammatory, anti-microbial, and anti-diabetic activities, among others (*Mcclements, 2019*), are limited because its low water solubility, a comprehensive evaluation of its solubility was undertaken. This evaluation encompassed seven distinct ChCl-based (Na)DES, all formulated in conjunction with sugars and polyols (*Jeliński, Przybyłek & Cysewski, 2019*).

The findings of the study revealed a significant enhancement in curcumin solubility within (Na)DES when compared to water. Notably, the ChCl-Glycerol (1:1) (Na)DES exhibited an exceptional increase in solubility, exceeding twelve thousand fold. Furthermore, it was observed that curcumin dissolved in ChCl-Glycerol (1:1) (Na)DES maintained its stability even when exposed to artificial sunlight, while the concentration of curcumin in MeOH dropped to only 5% of its initial amount after 120 min of exposure. These results confirm the protective impact of (Na)DES on molecules that are typically prone to degradation. They also illustrate that while the yield of curcumin obtained directly from plant sources using (Na)DES might be lower in comparison to organic solvents like MeOH, the former could offer a superior method for extending the shelf life of this bioactive metabolite. This is particularly interesting since (Na)DES could be considered as useful tools for the development of sustainable one-step extraction and safe storage of NP from natural sources. A similar study on the solubility and stabilization of curcumin in nineteen different (Na)DES showed that the best yields of curcumin were obtained using D-(+)-glucose-sucrose (1:1) and maleic acid-ChCl (1:3) (*Wikene, Bruzell & Tønnesen, 2015*). Further evaluation of both (Na)DES for their photo protecting capacity against solar radiation showed that the photostability of curcumin increased by 5.6 to ten-fold in D-(+)-glucose-sucrose (1:1) when compared to the same preparation in cyclodextrin, commonly used for drug delivery. These results can be explained by the capacity of the D(+)-glucose-sucrose (1:1) (Na)DES to form intermolecular hydrogen bonds which could lock the curcumin in one specific molecular conformation (keto form). Recently, the solubility of curcumin was investigated in hydrophobic (Na)DES using formulations based on camphor, menthol, and thymol in different molar ratios (1:1 to 1:5). In this case, it was observed that the solubility of curcumin is affected by both temperature and the molar ratio between the components of the hydrophobic (Na)DES (*Raja Sekharan et al., 2021*).

From this standpoint, (Na)DES have shown significant promise not only in enhancing the solubility of NP but also improving their bioavailability. The potential and

biocompatibility of (Na)DES as vehicles to improve the solubility and bioactivity of water-insoluble NP with pharmaceutical value were investigated using baicalin, curcumin, andrographolide, and oleanolic acid as model molecules. They showed a significant increase in the solubility of all molecules in (Na)DES when compared to that in water (*Cao et al., 2020*). Curcumin, oleanolic acid, baicalin, and andrographolide were found to be 250–4,000 times more soluble in (Na)DES than the control. The fact that the ChCl-leuvinic acid (1:2) was better for baicalin and curcumin, while lactic acid-1-propanol (1:1) increased the andrographolide solubility, and actamide:1-propanol (1:2) was the most appropriate for oleanolic acid, suggested that the NP solubility can be influenced by both their physicochemical characteristics and the (Na)DES composition. In order to establish the relationship between these factors, solvatochromic parameters such as polarity and the ability of (Na)DES to donate or accept hydrogen bonds ($\alpha$ and $\beta$ parameters), together with the electrostatic interactions between the solute and solvent (using the $\pi$ parameter) were evaluated. Interestingly, none of the solvatochromic parameters showed a clear relationship with the NP studied, indicating that the solvation ability of (Na)DES is a complex process. This is the result of a unique distinct bulk liquid and interfacial nanostructure of (Na)DES, which results from intra- and intermolecular interactions, including coulomb forces, hydrogen bonding, van der Waals interactions, electrostatics, dispersion forces, and apolar-polar segregation (*Bryant et al., 2022*).

## (Na)DES AS STORAGE FORMULATIONS

The reported capacity of (Na)DES for NP stabilization has suggested the use of these solvents as storage media (*Jeong et al., 2017*; *Jeliński, Przybyłek & Cysewski, 2019*; *Athanasiadis et al., 2018*). Consequently, several studies have delved into the degradation kinetics of NP, particularly those susceptible to oxidation, when stored in (Na)DES. One of the pioneering investigations scrutinized the stability of carthamin when stored in various (Na)DES at diverse temperatures and durations (*Dai, Verpoorte & Choi, 2014*). The results showed that carthamin remained stable after 7 days of storage in all solvents, including water and 40% EtOH (v/v) used as controls, when kept at −20 °C. However, after 15 days, curcumin's stability endured in ChCl-sucrose (4:1) (Na)DES, with less than 10% degradation in other (Na)DES compositions, while significant degradation occurred in the control samples. Notably, carthamin maintained its stability within the ChCl-sucrose (4:1) (Na)DES formulation for a month when stored at 4 °C and remained stable for at least three months at −20 °C. The stability of carthamin could be attributed to both viscosity and storage temperature; lower temperatures result in higher viscosity, facilitating stable molecular interactions with the (Na)DES.

Similarly, the capacity of ChCl-based (Na)DES evaluation to maintain salvianolic acid B (SAB) stable at room temperature for a period up to 60 days, showed that only 10% of SAB degraded when kept in ChCl based-(Na)DES for 24 days at room temperature. Besides, it remained almost without change after 60 days, while 90% of SAB degraded after 10 days when kept in water (*Chen et al., 2016*). FT-IR studies of the SAB degradation process suggested that (Na)DES inhibited ester hydrolysis in SAB, in addition to showing clear molecular interactions between SAB and (Na)DES.

In another investigation, the effectiveness of (Na)DES formulations, namely D-(+)-glucose-sucrose (1:1) and maleic acid-ChCl (1:2), in solubilizing curcumin was assessed. The results revealed that curcumin not only maintained its stability in both (Na)DES after being stored at room temperature for 1 month but also exhibited an increase in solubility. Specifically, the solubility of curcumin rose to 29 ± 4% in the case of D-(+)-glucose-sucrose (1:1) and to 16 ± 9% for maleic acid-ChCl (1:2) (*Wikene, Bruzell & Tønnesen, 2015*).

A similar study employed a chemometric approach to assess a variety of (Na)DES formulations, such as lactic acid-glucose (5:1), citric acid-glucose (1:1), and fructose-citric acid (1:1). The objective was to determine the optimal (Na)DES composition, along with the most effective temperature and water content, for the extraction of phenolic metabolites from specific plant materials. The outcomes unveiled that the highest extraction yield was achieved with the use of lactic acid-glucose (5:1) at 40 °C and 15% water (v/v) (*Fernández Mía de los et al., 2018*). Additionally, it has been reported that the presence of hydroxyl or carboxyl groups in the (Na)DES components can influence the stabilization of different metabolites through the formation of hydrogen bonds (*Meng et al., 2018*). In light of this, the study assessed the stabilization of eight phenolic compounds—namely oleuropein, hydroxytyrosol, coumaric acid, cinnamic acid, caffeic acid, ferulic acid, quercetin, and apigenin—in diverse (Na)DES formulations, stored at temperatures of 25 °C, 4 °C, and −18 °C.

The results unequivocally revealed that all phenolic metabolites retained their stability for a period exceeding two months when stored at the lowest temperature using the lactic acid-glucose (5:1) formulation. These findings confirm the great potential of (Na)DES for extraction, stabilization, and storage in NP studies.

More recently, two different studies investigated the use of (Na)DES as storage media for anthocyanins extracted from grape-pomace (*Panić et al., 2019*) and mulberries (*Bi et al., 2020*). Both studies evaluated the stability of these metabolites at room temperature (25 °C), 4°, and −18 °C for a period of up to 30 and 90 days, respectively. While (Na)DES formulations of ChCl-citric acid (2:1) and ChCl-proline-malic acid (1:1:1) resulted in better extraction of anthocyanins from grape pomace, the best yield of anthocyanins from mulberries was obtained using ChCl-lactic acid (1:2) hydrated with 20% of water. Both extracts were stored and used in degradation kinetics studies of anthocyanins, to determine the ability of (Na)DES formulations for anthocyanin-stability under storage conditions. The results obtained coincided with those reported previously (*Dai et al., 2016*; *Fernández Mía de los et al., 2018*), *i.e.*, a higher anthocyanin degradation was observed when both extracts were kept at room temperature, regardless of the solvent used. However, anthocyanin content in mulberry extracts was 1.7-fold higher in the (Na)DES formulation than in acidified EtOH, after three months of storage, whereas only 7% of the anthocyanins was lost when stored in (Na)DES at −20 °C. In the case of grape-pomace extracts, ChCl-citric acid (2:1) proved to have the highest stabilizing capacity at both 4° and −18 °C, after 60 days.

Compounds such as ascorbic acid are renowned for their robust antioxidant properties. Nevertheless, similar to anthocyanin, it is a highly labile compound, thus, *Gómez-Urios*

*et al. (2023)* studied the most suitable (Na)DES composition, including three sugar, one polyalcohol, and four carboxylic acids, for bioactive compounds extraction from orange peel. ChChl: malic acid (1:1) and ChChl:Glycerol (1:2) reported the best extraction yield than EtOH (50%) extracts. In addition, these (Na)DES combinations showed enhanced stability of the extract after 12 days of storage, specifically for ascorbic acid and hesperidin (majoritarian polyphenol in the orange peel).

The utilization of (Na)DES has also been explored for obtaining green tea (*Camellia sinensis*) extracts suitable for direct integration into cosmetic formulations (*Jeong et al., 2017*).

In this context, the need arises to preserve the stability of epigallocatechin-3-gallate (EGCG), the predominant catechin in green tea, which is susceptible to oxidation and epimerization. Ensuring the stability of EGCG is crucial for its practical application.

To address this, green tea extracts were prepared within a betaine-glycerol-glucose (4:20:1) (Na)DES, with a hydration level of 19% (w/w), and were compared to extracts in MeOH 70% (v/v), EtOH 100%, and 70% (v/v) EtOH. These extracts underwent thermal stress at 60 °C and were evaluated every 2 days over 3 weeks. The results confirmed the (Na)DES formulation as the most effective for stabilizing EGCG since 58% of EGCG could be detected after 3 weeks of storage, as compared to less than 39% detected in both MeOH and EtOH 70% (v/v). These results, coupled with the diverse array of components available for (Na)DES formulations, suggest that these solvents have the potential to serve as efficient mediums for the extraction and preservation of various catechins derived from tea.

In a similar study focusing on the utilization of olive leaves, which are esteemed for their high polyphenolic content and are a notable by-product of the food industry, the study sought to assess the long-term storage stability of olive leaf extracts. To achieve this, they compared a glycerol-glycine-water (7:1:3) (Na)DES formulation to the same (Na)DES formulation enriched with 9% (w/v) of methyl-β-cyclodextrin (m-β-CD). These were compared to an EtOH 60% (v/v) control when stored at temperatures of 4 °C, 22 °C, and 55 °C for 20 days. The results showed that the glycerol-glycine-water had a stabilizing effect on the extracts at all temperatures and that the incorporation of m-β-CD to the extracting (Na)DES resulted in a significant decrease in the oxidation of polyphenolics. Even though the exact mechanism of stabilization is unclear, m-β-CD may form a complex with phenolic metabolites through intramolecular hydrogen bonding (*Athanasiadis et al., 2018*).

Another molecule susceptible to various physicochemical influences is β-carotene, a natural pigment known for its insolubility in water and limited solubility in both oils and organic solvents. *Basar et al. (2020)* investigated the use of (Na)DES to increase the stability and loading of β-carotene in whey protein concentrate capsules prepared using emulsion electrospraying. Their investigation encompassed four ChCl-based (Na)DES formulations, with β-carotene serving as the dispersed phase and whey protein concentrate as the continuous phase. The outcomes highlighted that the most effective solubilization of β-carotene was achieved with a ChCl-butanediol (Na)DES formulation. Furthermore, a combination of this (Na)DES in a 10:90 ratio with whey protein concentrate led to the

creation of stable emulsions. Additionally, the loading capacity in these formulations was on par with that obtained using suspensions of β-carotene particles in glycerol.

As observed previously, the encapsulation of these nanoparticles also exhibited a protective effect against oxidation.

## (Na)DES TOWARDS THE DEVELOPMENT OF READY-TO-USE FORMULATIONS

Although the solvation and stabilization properties of (Na)DES on a wide range of NP offer a new perspective towards the storing of NP, the possibility of keeping the NP, and besides that can preserve its biological activity offers the possibility to develop (Na)DES-based formulations ready-to-use.

In a related study, Mouratoglou, Malliou & Makris (2016) conducted a comparative analysis of the antioxidant activity in extracts derived from agricultural and food waste. They compared the use of water and 60% EtOH (v/v) with several glycerol-based (Na)DES formulations. The findings demonstrated that both the extracts' yield and the antioxidant activity levels were significantly higher when employing glycerol-based (Na)DES as compared to water or aqueous EtOH. Additionally, an observed positive correlation between ORAC antioxidant activity and the total phenolic and anthocyanin content was noted when evaluating five ChCl-based (Na)DES (Radošević et al., 2016). The enhanced extraction efficiency and increased antioxidant activity observed with the utilization of a ChCl-malic acid (1:1) formulation indicate that (Na)DES have the potential to elevate antioxidant activity. This phenomenon can likely be attributed to the inherent antioxidant properties of one or more components within the (Na)DES formulation. Moreover, it is conceivable that a synergistic effect between the phenolic compounds and the (Na)DES components contributes to this heightened antioxidant activity. These findings coincided with those reported by Mitar et al. (2019) on the antioxidant effect of eight (Na)DES based on organic acids and sugars. In this case, the evaluation of the antioxidant activity of ChCl-malic acid (1:1), proline-malic acid (1:1), ChCl-proline-malic acid (1:1:1), betaine-malic acid (1:1), malic acid-glucose (1:1), malic acid-glucose-glycerol (1:1:1), ChCl-citric acid (1:1), and betaine-citric acid (1:1) showed that the lowest antioxidant activity was observed in (Na)DES formulations containing citric acid when compared to those containing malic acid. These results can be easily elucidated by the fact that malic acid exhibits a superior antioxidant activity in comparison to citric acid.

Although the antioxidant activity of some extracts could be attributed to the composition of the (Na)DES, it is important to note that it can also be influenced by the phytochemical profile of each extract obtained in each case. On the other hand, the antioxidant mechanisms of either the extracts contained in the (Na)DES or themselves are still unclear and may be due to both physical and chemical contributions. Besides the mixture effects resulting in synergism or antagonism amongst the various polyphenols and/or (Na)DES occurring in the extracts could not be ruled out (Mouratoglou, Malliou & Makris, 2016).

A comparable assessment of the antioxidant and antimicrobial effects of ten (Na)DES formulations, encompassing ChCl, betaine, and citric acid as HBAs, along with different

HBDs (such as organic acids, sugars, sugar alcohols, amino acids, and amides), revealed that (Na)DES lacking antioxidant species like urea, xylitol, and sorbitol, did not show any antioxidant activity (*Radošević et al., 2018*). However, (Na)DES formulated with antioxidant components (*e.g.*, organic acids) were consequently antioxidant, as previously reported.

Interestingly, all (Na)DES, except for ChCl-xylitol (5:2), betaine-glucose (5:2), and ChCl-sorbitol (2:3), inhibited the growth of some microorganisms (*e.g.*, *Escherichia coli, Proteus mirabilis, Salmonella Typhimurium, Pseudomonas aeruginosa, Staphylococcus aureus*).

Because of these findings, the antimicrobial effect of NP extracts obtained using (Na)DES was evaluated to explore the possibility of using them directly as part of cosmetic and/or pharmaceutical formulations (*Grozdanova et al., 2020*). While the best extraction results of total phenolics were obtained using ChCl-glucose (5:2) containing 30% water, none of the extracts were active against all tested microorganisms. However, citric acid-1,2 propanediol (1:4) extracts showed the best antimicrobial activity against S. *pyogenes, E. coli, S. aureus, and C. Albicans.*

Due to the extensive medicinal properties associated with *Moringa oleifera*, a recent study examined the antioxidant and antibacterial activities of flavonoids extracted from its leaves using (Na)DES, specifically ChCl-Urea (1:2). The assessment of antioxidant capacity was conducted through DPPH, ABTS, FRAP, and ORAC assays. The results demonstrated that the (Na)DES-extracted *Moringa oleifera* leaves exhibited superior activity in comparison to the positive control (Vc) in the case of DPPH, ABTS, and ORAC assays. Notably, for the FRAP assay, the (Na)DES-extracted sample outperformed those obtained using ethyl acetate. Moreover, the investigation explored the antimicrobial potential by employing a bacterial growth inhibition method. The inhibitory effect of the (Na)DES-extracted sample was evaluated against five bacterial strains, with minimal inhibitory concentrations (MICs) of 1.25 mg/mL for *E. coli, Staphylococcus aureus, Bacillus subtilis*, and *Pseudomonas aeruginosa*, and 0.625 mg/mL for *Proteus common*. These MIC values were twice as low as those observed for propyl p-hydroxybenzoate (used as a control). Furthermore, within the same study, the authors investigated the microencapsulation of the (Na)DES-based extract from *Moringa oleifera* leaves to assess flavonoid release and bioavailability. The microcapsules were prepared using xylose-modified soybean protein isolate and gelatin as wall materials, and simulated *in vitro* digestion of the (Na)DES-based extract microcapsules was examined. The results revealed that the intestinal release rate of (Na)DES-based extract microcapsules was 86 times higher than that of the free (Na)DES-based extract (*Wei et al., 2023*). Since microcapsules have been widely used in the field of food, those results could be key for the research toward the development of ready-to-use in (Na)DES formulations.

The extraction of rhizomes, leaves, and flowers of *Curcuma longa L* using (Na)DES-based menthol and ChCl (*Oliveira et al., 2021*) showed that the best yields were obtained when using menthol-lactic acid (1:2) and menthol-acetic acid (1:1) as solvents. The antibacterial activity evaluation using the disk diffusion method showed that the leaf extracts obtained with menthol-acetic acid (1:1) had a comparable activity with that of the

antibiotic used as a positive control. *Listeria monocytogenes* proved to be the most sensitive to all menthol-lactic acid (1:2) extracts, mostly from leaves and flowers, suggesting a possible synergistic effect between menthol and lactic acid, since both are known to have an antimicrobial effect. Additionally, the iron-chelating capacity of the extracts was evaluated to explore the possibility of their being used as food additives; the results showed that pure (Na)DES such as menthol-lactic acid (2:1) and menthol-acetic acid (1:1) had the highest activity, with values above 90% of chelated $Fe^{2+}$. The fact that the extracts of menthol-lactic acid (2:1) exhibited a significantly superior result when compared to that of menthol-acetic acid (1:1), is likely due to the lower binding capacity of (Na)DES because of intermolecular binding with NP which occupy chelating sites of the solvent, which consequently reduces its iron-chelating ability.

Recently, the potential of (Na)DES in fortified food formulations was explored through an assessment of the antioxidant activity in ChCl and betaine-based (Na)DES extracts derived from cocoa by-products (*Manuela et al., 2020*). Initial assessments of the phenolic content within different (Na)DES extracts revealed that (Na)DES formulations incorporating glucose as a HBD yielded the most favorable results in comparison to the reference, which was an extract in 70% EtOH (v/v). Among the various solvents evaluated, betaine-glucose (1:1) demonstrated the highest antioxidant activity. In contrast, ChCl-glucose (1:1), ChCl-citric acid (2:1), and 70% EtOH (v/v) displayed similar levels of activity. While previous studies have highlighted a correlation between antioxidant activity and the total phenolic and procyanidin content (*Radošević et al., 2016*), no such correlation was observed between antioxidant activity and the levels of catechin, protocatechuic acid, and epicatechin. These outcomes suggest that the antioxidant properties of an extract may arise from the synergistic effects of various types of polyphenols rather than being attributed to specific polyphenolic compounds (*Manuela et al., 2020*). Although the use of (Na)DES, both in formulation media and to enhance the biological activity of NP still requires further studies, this investigation certainly brings this possibility closer.

Most recently, the suitability of Na(DES) to extract lipids and pigments from spirulina has been studied, together with the impact of (Na)DES formulations on bacterial strains related to the skin microbiome and inflammation (*Wils et al., 2021*). Since the spirulina-producing cyanobacteria are also sources of bioactive metabolites such as carotenoids, free fatty acids (FFA) and, especially, polyunsaturated fatty acids (PFA), different (Na)DES formulations containing hydrophilic and lipophilic components were selected, *e.g.*, glucose-glycerol-$H_2O$ (1:2:4), betaine-glycerol (1:2) and (1:8), octanoic acid-lauric acid (3:1), nonanoic acid-decanoic acid-lauric acid (3:2:1), and menthol-levulinic acid (1:2). Both the (Na)DES extracts and the (Na)DES themselves were evaluated in terms of their effect on inflammation and bacterial skin microbiota. While the glucose-glycerol-$H_2O$ (1:2:4) (Na)DES extract reduced the release of pro-inflammatory mediators, a similar effect was not observed when the (Na)DES alone was tested, suggesting that the anti-inflammatory effect could be due to the phycocyanin content in the (Na)DES extract. A similar evaluation of the antibacterial effect of the (Na)DES extracts and the control formulations on the viability of *Corynebacterium xerosis*, *Staphylococcus epidermidis, Cutibacterium acnes*, and *S. aureus* showed that lipophilic

(Na)DES-based on medium-chain fatty acid exhibited antibacterial activity against all strains, with *S. aureus* and *S. epidermidis* being the most sensitive. However, a similar bactericidal effect observed in the (Na)DES controls suggested that their components were mainly responsible for the antibacterial activity. Interestingly, and even though that menthol is well recognized for its antimicrobial property against human and plant pathogens (*Iscan et al., 2002*; *Mbous et al., 2017b*), the antibacterial activity of the menthol-levulinic acid 1:2 (Na)DES formulation was lower.

Although neither the extracts using hydrophilic (Na)DES formulations nor the hydrophilic (Na)DES controls, showed significant antibacterial activity. This investigation represents the first step in the potential use of (Na)DES as a functional ingredient in cosmetic formulations.

Regarding this, *Akbar et al. (2023)* examined the antibacterial properties of eleven different (Na)DES-based in methyl-trioctylammonium chloride as a hydrogen-bond acceptor (HBA). These (Na)DES were tested for their bactericidal activities against Gram-positive (*Streptococcus pyogenes, Bacillus cereus, Streptococcus pneumoniae*, and *Staphylococcus aureus*) and Gram-negative (*Escherichia coli K1, Klebsiella pneumoniae, Pseudomonas aeruginosa*, and *Serratia marcescens*) bacteria. The results indicated that named DES-4 (methyl-trioctylammonium chloride-glycerol 1:1) and DES-11 (methyl-trioctylammonium chloride-fructose 1:1.25) exhibited broad-spectrum antibacterial behavior and the highest bactericidal activity at a 2 μL dose. DES-4 displayed significant antibacterial effects against *Pseudomonas aeruginosa* and *E. coli* K1, while DES-11 effectively eliminated of *E. coli* K1 and *P. aeruginosa*. Among Gram-positive bacteria, DES-4 inhibited *Bacillus cereus* and *Streptococcus pneumoniae*, while DES-11 displayed bactericidal effects against *B. cereus* and S. *pneumoniae*. Furthermore, the biocompatibility of the (Na)DES was evaluated against HeLa cells, which showed minimal cytotoxicity towards human cells, except for DES-10 (methyltrioctylammonium chloride-citric acid 1:1), which exhibited notable cytotoxic effects. These findings suggest that the tested (Na) DES hold potential as novel antibacterial drugs for combating bacterial infections. However, it is important to emphasize that the cytotoxicity profile of (Na)DES is closely related to its composition. For example, some studies have revealed that (Na)DES based on organic acids tends to be more toxic. Similarly, it has been suggested that the longer the chain of the functional groups composing these solvents, the more toxic they are (*Kudłak, Owczarek & Namieśnik, 2015*; *Hayyan et al., 2016*).

In a study focused on innovative formulation media, (Na)DES were investigated as a novel medium for ocular formulations. Multiple (Na)DES systems, comprised of combinations of sugars, polyols, amino acids, and choline derivatives, were assessed to enhance the solubility of acetazolamide, a drug known for its limited solubility and use in glaucoma treatment. To ensure the biocompatibility of (Na)DES, cytotoxicity assays were also conducted. The results indicated that betaine-N-acetyl cysteine (1:1) and betaine-Glycerol (1:2) emerged as the most promising systems due to their favorable cytotoxicity profiles and their capacity to enhance the solubility of acetazolamide. The authors suggested that these findings have the potential to advance the development of ocular drug formulations, particularly in the form of eye drops, by extending drug

residence times and improving bioavailability (*Sarmento et al., 2023*). The approach used in this study is undoubtedly enriching to guide other studies in the integration of NP into (Na)DES formulations where the molecules of interest remain active and can be used in various applications, such as the pharmaceutical or food industry.

## (Na)DES AS A TOOL IN SOLUBILITY AND PERMEABILITY OF ACTIVE PHARMACEUTICAL INGREDIENTS IMPROVEMENT

The ability of (Na)DES to establish a complex network of non-covalent bonds between its components and the NP target has been used to explain their capacity to dissolve a wide range of molecules. This ability has been explored to improve the poor water solubility of some active pharmaceutical ingredients (API), although it has also been reported that some APIs can themselves act as components of a (Na)DES, resulting in an enhancement of their solubility, skin permeation, and absorption (*Stott, Williams & Barry, 1998*).

More recently, the advancement of bioactive eutectic systems incorporating APIs as integral components of (Na)DES has expanded the potential utility of these systems within pharmacy and biomedicine (*Pedro et al., 2019*). This innovation offers the prospect of creating drug delivery vehicles for enhanced stability and activity of APIs, while simultaneously inhibiting their crystallization upon dispersion in aqueous media.

*Aroso et al. (2016)* were pioneers in introducing the concept of Therapeutic Deep Eutectic Solvents (THEDES). They explored ChCl or menthol-based THEDES in conjunction with three different APIs: acetylsalicylic acid, benzoic acid, and phenylacetic acid. Their objective was to investigate the feasibility of creating a liquid formulation that could enhance both the bioavailability and delivery rate of these APIs. The outcomes of their research demonstrated that the dissolution of APIs within THEDES in a buffer occurred significantly faster than the dissolution of the APIs in their solid form. For instance, the dissolution rate of an individual API in the menthol-based THEDES was notably faster (15–45 min) than when the API was used alone (>24 h). This underscores that, much like the dissolution of nanoparticles is influenced by the selection of the Hydrogen Bond Acceptor (HBA), the dissolution of APIs can be tailored by selecting the nature and ratio of components in a given (Na)DES formulation.

Moreover, the menthol-based THEDES exhibited heightened antimicrobial activity when evaluated against both Gram-positive bacteria (*Bacillus subtilis* and *Staphylococcus aureus*) and Gram-negative bacteria (*E. coli*) in comparison to the pure API. This finding underscored the potential of (Na)DES to enhance the antimicrobial activity of specific APIs.

Similarly, evaluation of the solubility and permeability of APIs in menthol-based systems demonstrated that the resulting THEDES formulations not only enhanced drug solubility in an isotonic solution at pH 7.4 but also presented an enhanced permeability (*Duarte et al., 2017*). The highest increase (12 fold) in solubility was observed for a menthol-Ibuprofen (3:1) system, while the API solubility in menthol-benzoic acid and menthol-phenylacetic acid increased two-fold. The menthol-ibuprofen THEDES showed the best permeability results. It is important to mention that, although ibuprofen is classified as a Class III compound with high solubility but low permeability in the

biopharmaceutical classification, when incorporated as an API in the THEDES system, it is reclassified as a Class I system, exhibing high solubility and permeability.

In addition to enhancing the solubility of different APIs, THEDES systems have also been considered as new alternatives to minimize the side-effects and drug resistance of conventional medicaments, *e.g.*, pharmaceuticals used for the treatment of tuberculosis. This treatment tipically require prolonged regimen of multiple antimycobacterial drugs, causing various secondary effects (*Aroso et al., 2016*). An investigation (*Santos, Leitão & Duarte, 2019*) of THEDES formulations designed to enhance the properties of ethambutol and L-arginine showed that THEDES prepared using citric acid as the HBA dramatically increased the solubility of ethambutol to 27.5 fold and its permeability to 1.5 fold, which significantly improved its bioavailability. These findings one again confirm that the incorporation of APIs to THEDES systems represents a valuable alternative to improve the solubility and permeability properties of bioactive products.

In a different study, a series of limonene-based THEDES were prepared to investigate the potential of these systems on cancer therapy. Limonene is well recognized for its chemo-preventive and chemotherapeutic effects on several types of cancer (*Pereira et al., 2019*). However, since limonene is highly toxic and often compromises cell viability, its toxicity in different THEDES was explored. The results indicated that while all THEDES formulations of limonene with myristic acid, menthol, capric acid, and ibuprofen display antiproliferative properties, only the ibuprofen-limonene (1:4) system was able to inhibit cell proliferation without comprising cell viability. Additionally, the evaluation of the potential of limonene-based THEDES to increase the solubility of ibuprofen showed that ibuprofen-limonene 1:4 and 1:8 systems resulted in 4.32 and 5.63-fold solubility increase, respectively, when compared to the solubility of ibuprofen in powder form (*Pereira et al., 2019*). These findings highlight the potential synergistic role played by the (Na)DES components, which was confirmed by the fact that limonene toxicity decreased when it was part of the THEDES system. These findings are promising for the development of new drug delivery systems capable of improving API solubility and cytotoxicity.

The hydrotropy effect of THEDES has been demonstrated in several studies. For instance, THEDES formulations with fluconazole and mometasone showed that both drugs improved their solubility until 120-fold in all the tested capric acid/menthol mixtures. Additionally, these THEDES formulations were used to prepare a topical cream, ad its clinical compatibility was tested on Albino rabbits (*Staufflan, White* strain) whit no side effects were observed (*Mbous et al., 2017b*).

The use of NP in THEDES systems formulations undeniably offers a vast range of applications within the pharmaceutical sector. However, it is imperative to acknowledge that the behavior and mechanism of action of THEDES are profoundly influenced by the types of components used, and the associations established between them. While THEDES can offer advantages such as improved drug dissolution, enhanced drug penetration, and acting as a synthetic medium for drug carriers, their potential applications as a universal drug delivery system not limited to any particular type of drug require further exploration. Therefore, additional studies are essential to fully understand and unlock the potential of THEDES systems in drug delivery applications.

## CONCLUSIONS AND PERSPECTIVES

This review shows that, nearly almost two decades after the first description of (Na)DES, there has been a rapid expansion in the number of publications focused on the applications of these solvents. In the context of NP research, the use of (Na)DES has taken advantage of their tunable characteristics directed towards a target compound. This feature has allowed the use of (Na)DES as "green" extraction solvents with yields that are comparable, or even better, to those obtained by conventional solvents. However, this characteristic is often questionable, due to the large number of (Na)DES reported in the literature, especially those based on choline or urea (both synthetically produced), which do not meet the requirements for being considered sustainable.

On the other hand, while most investigations are limited to demonstrate their efficiency and listing their advantages over conventional solvents, several studies have also indicated the drawbacks of using (Na)DES as extracting solvents, including their low volatility, the possibility to dissolve non-target molecules, and the difficulty to recover the molecule of interest from eutectic mixtures. These factors still represent an important limitation for the application of these systems in obtaining NP from plant matrices on an industrial scale. Therefore, the true advantage to use (Na)DES in the NP investigation goes beyond their simple use as an extraction solvent. Instead, they should be viewed from a perspective where these solvents could be integrated into the desired final product, taking advantage of their unique characteristics. In this sense, some studies mentioned in this document have been carried out with this goal, revealing that (Na)DES could increase the useful life of NPs in different environmental conditions. Additionally, they can be used for the transport of APIs, offering advantages such as the extension of its useful life and the improvement of its pharmacokinetics.

Currently, several commercial enterprises are exploring the industrial applications of (Na)DES. The special features of (Na)DES that can aid in stabilizing, solubilizing, and formulating media where the biological properties of NP are not only maintained but also improved, represent an advantage that deserves to be taken into account.

### Funding

This work was supported with a postdoctoral fellowship financing financed by The National Council for Science and Technology (CONACYT, Mexico). The funders had no role in study design, data collection and analysis, decision to publish, or preparation of the manuscript.

### Grant Disclosures

The following grant information was disclosed by the authors:
The National Council for Science and Technology (CONACYT, Mexico).

### Competing Interests

The authors declare that they have no competing interests.

## Author Contributions

- Mariana Ruesgas Ramon conceived and designed the experiments, performed the experiments, analyzed the data, performed the computation work, authored or reviewed drafts of the article, and approved the final draft.
- Erwann Durand analyzed the data, authored or reviewed drafts of the article, contributed recent literature on the topic to enhance the review, and recommended journals for its publication, and approved the final draft.
- Karlina Garcia-Sosa performed the computation work, prepared figures and/or tables, and approved the final draft.
- Luis Manuel Peña-Rodríguez analyzed the data, authored or reviewed drafts of the article, and approved the final draft.

## Data Availability

This is a literature review.

## Supplemental Information

Supplemental information for this article can be found online at http://dx.doi.org/10.7717/peerj-achem.28#supplemental-information.

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
