# Peer review of "Exploring the potential of deep eutectic solvents (DES) in bioactive natural product research: from DES to NaDES, THEDES, and beyond"

_PeerJ Analytical Chemistry, doi:10.7717/peerj-achem.28_

## Round 0.1 · original submission · Minor Revisions

Dear Authors,

Carry out the revisions requested by the reviewers on your submission.

Reviewers 1 & 2 have requested that you cite specific references. You may add them if you believe they are especially relevant. However, I do not expect you to include these citations, and if you do not include them, this will not influence my decision.

**Language Note:** PeerJ staff have identified that the English language needs to be improved. When you prepare your next revision, please either (i) have a colleague who is proficient in English and familiar with the subject matter review your manuscript, or (ii) contact a professional editing service to review your manuscript. PeerJ can provide language editing services - you can contact us at copyediting@peerj.com for pricing (be sure to provide your manuscript number and title). – PeerJ Staff

Reviewer 1 ·

Basic reporting

no comment

Experimental design

no comment

Validity of the findings

no comment

Additional comments

no comment

Annotated reviews are not available for download in order to protect the identity of reviewers who chose to remain anonymous.

Reviewer 2 ·

Basic reporting

Below are the comments on the manuscript entitled “Exploring the Potential of Deep Eutectic Solvents (DES) in Bioactive Natural Product Research: From DES to NaDES, THEDES, and Beyond”.
General comments:
The review article provides a comprehensive insight into NaDES, extraction methodologies of functional compounds, and their possible applications. The novelty of the manuscript (MS) is fair.

Experimental design

The manuscript is appealing; however, in recent years, many reviews have been published on the subject that should be included:
Saini, R., Kumar, S., Sharma, A., Kumar, V., Sharma, R., Janghu, S., & Suthar, P. (2022). Deep eutectic solvents: The new generation sustainable and safe extraction systems for bioactive compounds in the agri-food sector: An update. Journal of Food Processing and Preservation, 46(10), e16250. https://doi.org/10.1111/jfpp.16250
Suthar, P., Kaushal, M., Vaidya, D., Thakur, M., Chauhan, P., Angmo, D., Kashyup, S. & Negi, N. (2023). Deep eutectic solvents (DES): An update on the applications in food sectors. Journal of Agriculture and Food Research, 100678. https://doi.org/10.1016/j.jafr.2023.100678
Vast work has been already done on extraction from DES and its applications. Justify the novelty of the work in the abstract.
There are some minor syntax and grammatical errors in the text though the MS was generally understandable. Also, typing errors appear several times. e,g., line no. 87 Prowess must be written as a process.
Section 3:
Check the citation style throughout the MS. e.g., Line. 147: Citation style is not appropriate
Check all the abbreviations used. Abbreviations used in the whole manuscript have to be defined first and then their abbreviations have to be used. e.g., Line 153 please provide the abbreviation used for IL.
Please check the references again for inconsistencies according to the format of the journal. All references cited in the manuscript should be included in the Reference list, and all references included in the Reference List should be cited in-text.
Check for the consistency of the font size of the text in Table 1
Please use high-quality structures for Figure 1 to improve the clarity.

Validity of the findings

The review article provides a comprehensive insight into NaDES and its novel applications. Therefore, this MS is interesting and it is worthy of publication in terms of scientific and technical merit. Although there are some concerns/amendments that need to be carried out, it can be suitable for publication after minor revisions.

Reviewer 3 ·

Basic reporting

This review is of paramount importance considering the increasing interest in utilizing Deep Eutectic Solvents (DES) and (Na)DES in the field of NP (Nanoparticle) research. While existing reviews predominantly focus on their extraction capabilities, this review takes a broader perspective, delving into their potential applications beyond extraction. By consolidating findings that go beyond mere extraction, this manuscript highlights the versatility of DES and (Na)DES, encouraging researchers to explore their novel uses. Given the growing emphasis on sustainable and innovative solutions in the scientific community, this review bridges the gap between the narrow, extraction-centric viewpoint and the numerous possibilities that these solvents offer.

Furthermore, this review is tailored for researchers, scientists, and professionals actively involved in NP research, innovative technologies, bio-formulations, and related fields. It will serve as a valuable resource for those seeking to broaden their understanding of DES and (Na)DES, extending beyond their conventional uses. Additionally, this review will be beneficial to individuals interested in sustainable chemistry and the development of innovative solutions for both industrial and biomedical challenges. Its insights provide information on current advancements and open up new avenues for research, making it pertinent across academic and industrial sectors.

Experimental design

The study design appears to be well-conceived, with a clear research question and a structured approach to gathering, analyzing, and categorizing relevant literature. It should provide valuable insights into the applications of DES in bioactive natural product research.

Validity of the findings

The conclusions drawn in the review are valid, as they are based on a critical analysis of the literature and are supported by existing scientific knowledge and research. The review contributes to expanding the understanding of the diverse applications of DES in natural product research and hints at promising avenues for future advancements in this field. However, it's important to note that the validity of these findings may evolve as further research and experimentation are conducted in this area.

Additional comments

I recommend this paper for publication due to its comprehensive coverage of an emerging and interdisciplinary topic, which can greatly benefit researchers and professionals in various fields.

---

## Round 0.2 · accepted · Accept

The authors have revised the manuscript following the reviewer's suggestions. I believe that your manuscript is now polished into a version that is publishable in a journal with an international audience.